

# PKC in motorneurons underlies self-learning, a form of motor learning in *Drosophila*

Julien Colomb and Björn Brembs

Biologie, Chemie, Pharmazie, Institut für Biologie-Neurobiologie, Freie Universität Berlin, Berlin, Germany
Institute of Zoology—Neurogenetics, Universität Regensburg, Regensburg, Germany

## ABSTRACT

Tethering a fly for stationary flight allows for exquisite control of its sensory input, such as visual or olfactory stimuli or a punishing infrared laser beam. A torque meter measures the turning attempts of the tethered fly around its vertical body axis. By punishing, say, left turning attempts (in a homogeneous environment), one can train a fly to restrict its behaviour to right turning attempts. It was recently discovered that this form of operant conditioning (called operant self-learning), may constitute a form of motor learning in *Drosophila*. Previous work had shown that Protein Kinase C (PKC) and the transcription factor *dFoxP* were specifically involved in self-learning, but not in other forms of learning. These molecules are specifically involved in various forms of motor learning in other animals, such as compulsive biting in *Aplysia*, song-learning in birds, procedural learning in mice or language acquisition in humans. Here we describe our efforts to decipher which PKC gene is involved in self-learning in *Drosophila*. We also provide evidence that motorneurons may be one part of the neuronal network modified during self-learning experiments. The collected evidence is reminiscent of one of the simplest, clinically relevant forms of motor learning in humans, operant reflex conditioning, which also relies on motorneuron plasticity.

## INTRODUCTION

Motor learning is a loosely defined term that mainly encompasses motor adaptation and skill acquisition (*Kitago & Krakauer, 2013*). Different nervous system areas, from the spinal cord, through the cerebellum and the basal ganglia to the motor cortex, were shown to be involved in different types of motor learning (see *Kitago & Krakauer (2013)* for a review). Conceptually and biologically, skill learning has also been related to habit formation (*Graybiel & Grafton, 2015*; *Santos et al., 2015*; *Jin & Costa, 2015*; *Costa, 2011*). In a first phase, variability of motor actions is generated, probably actively as in zebra finch song leaning (*Woolley & Kao, 2015*). Then, any mismatch between the predicted and the observed outcome of the action provides an error signal, changing the properties of the neural networks involved in controlling the behavior. Commonly, this procedure serves to reduce the amount of variability in the movements, but it can also increase variability (*Neuringer, 2002*).

Corresponding author
Björn Brembs, bjoern@brembs.net

One of the simplest cases of motor learning may be operant conditioning of spinal cord reflexes, which was studied in mice, rats, macaques and humans (*Wolpaw & Lee, 1989*; *Wolpaw, 2010*). In this experiment, the variability of spinal reflex amplitudes is reduced to lower (or higher, respectively) than baseline values by rewarding only those reflexes with amplitudes lower (or higher, respectively) than the pre-training baseline. This procedure is currently being used to treat patients with spinal cord injury (*Thompson, Pomerantz & Wolpaw, 2013*). In *Aplysia* operant self-learning, rewarding initially highly variable biting behavior with food, increases the probability of spontaneous biting attempts, until biting becomes very repetitive and essentially compulsive (*Nargeot, Bon-Jego & Simmers, 2009*; *Bedecarrats et al., 2013*; *Sieling et al., 2014*; *Nargeot, Petrissans & Simmers, 2007*). Both forms of motor learning seem to involve dopamine-induced intrinsic plasticity in motor and pre-motor neurons in vertebrates (*Wolpaw, 2010*) and *Aplysia* (*Lorenzetti, Baxter & Byrne, 2008*; *Nargeot, Petrissans & Simmers, 2007*). Protein kinase C (PKC) (*Carp & Wolpaw, 1994*; *Lorenzetti, Baxter & Byrne, 2008*; *Watanabe et al., 2006*) and FoxP2 (*Lai et al., 2001*; *Fee & Scharff, 2010*) are two known molecular players in motor learning. Elegant experiments in *Aplysia* demonstrated how convergence of contingent behavior-mediated PKC activity and reward-mediated, dopamine-dependent PKA activation in a decision-making neuron accomplished motor learning in this model (*Lorenzetti, Baxter & Byrne, 2008*).

Despite pioneering work carried out in insects in the 1970s and 80s (e.g., *Hoyle, 1979*), there are currently only few invertebrate motor learning paradigms in use. The fruit fly self-learning experiment (originally devised by *Wolf & Heisenberg, 1991*), in *Drosophila melanogaster* was shown to depend on PKC (*Brembs & Plendl, 2008*) and the *Drosophila* orthologue of the human FOXP2 gene, *dFoxP* (*Mendoza et al., 2014*). Self-learning also seems to be related to habit formation (*Brembs, 2009*; *Mendoza et al., 2014*), such that its classification as a form of motor learning appears straightforward. In this experiment, a fly is glued to a hook and flies stationary in a homogeneous environment, tethered to a torque meter that measures the force of its turning attempts around its vertical body axis (yaw torque). After 8 min of training, where one of the turning directions is systematically punished via an infrared laser beam, wild type *Drosophila* restrict their behavior to the previously safe turning directions. Although developed to study operant conditioning (*Wolf & Heisenberg, 1991*) and its two components self- and world-learning (*Colomb & Brembs, 2010*), the experiment can be conceptualized in the framework of motor learning: The variability in spontaneous turning attempts (*Maye et al., 2007*) becomes restricted by learning.

With its powerful genetic toolbox, the *Drosophila melanogaster* model system is particularly well-suited to study both the molecular processes and the neuronal circuits involved in motor learning. It was previously found that ubiquitous PKC inhibition (*Brembs & Plendl, 2008*), as well as mutations in the *dFoxP* gene (*Mendoza et al., 2014*), specifically affect self-learning in *Drosophila melanogaster*. Here, we present our attempts at discovering which PKC gene is responsible for the defect, before presenting experiments strongly suggesting that the action of PKC is required in motorneurons during self-learning at the torque meter.

## METHODS

### Fly care

All flies were kept on standard cornmeal/molasses medium (*Guo et al., 1996*) at 25 °C and 60% humidity with a 12 h light/12 h dark regime (*Brembs, 2008*). We obtained most lines from the Bloomington *Drosophila* stock center: FBti0077657 (PKC53e), FBti0041429(PKCinac), FBti0041818 (PKCdelta) were outcrossed in the WTB background for 6 generations; tub-Gal80$^{ts}$ (FBtp0002650 on the second chromosome, insertion identifier lost), d42-Gal4 (FBti0002759) and elav-Gal4 (FBti0002575) were outcrossed to our CS background for 6 generations, before being crossed to their respective effector lines. The UAS-PKCi (FBti0010565) line, kindly provided by Henrike Scholz, was outcrossed for 6 generations into a CS background before being combined with tub-Gal80$^{ts}$. Other lines were not outcrossed and obtained from the VDRC or from colleagues: FBti0119252 (PKCinacRNAi) and FBti0159711(PKC-53eRNAi) were obtained from the VDRC; the 7y-c819 Gal4 double line (FBti0015361 and FBti0018454) was provided by Jean-René Martin, c232-Gal4 (FBti0002929) by Roland Strauss, H24-Gal4 (FBti0016292) by Martin Heisenberg, c380-Gal4 (FBti0016294) by Stephan Sigrist and the d42Gal4,chaGal80 double line (unknown identifiers) by Carsten Duch. After briefly immobilizing 24–48 h old female (kept in mixed gender population until then) flies by cold-anesthesia, head and thorax were glued (Sinfony Indirect Lab Composite; M ESPE, St. Paul, MN, USA) to a triangle-shaped copper hook (diameter 0.05 mm) the day before the experiment. The animals were then kept individually in small moist chambers containing a few grains of sucrose until the experiment. For the UAS-PKCi related experiments, flies were grown at 25 °C and received a heat shock (35 °C for 4 h) 0.5–4 h before being tested at 25 °C, if not noted otherwise. For RNAi experiments, flies were grown at 25 °C and kept for two days at 32 °C after eclosion before being tested at 25 °C.

### Torque meter setup

The core device of the set-up is the torque meter, as described elsewhere (*Tang & Juusola, 2010*). It measures a fly's angular momentum around its vertical body axis, caused by intended flight maneuvers (yaw torque). The fly, glued to the hook as described above, was attached to the torque meter via a clamp to accomplish stationary flight in the center of a cylindrical panorama (arena, diameter 58 mm), which was homogeneously illuminated from behind. The light source was a 100 W, 12 V tungsten-iodine bulb. The arena was illuminated with 'daylight' by passing it through a blue–green filter (Rosco 'surfblue' No. 5433). An analogue to digital converter card (ADC-USB-120FS, measurement computing Inc.; Norton, MA, USA) which transformed the analog signal into a 12 bit digital signal that feeds into the computer, and software to control the experiment and record the data (LabView, National Instruments Germany GmbH, Ganghoferstrasse, Muenchen, Germany). Punishment was achieved by applying heat from an adjustable infrared laser (StockerYale Lasiris SNF series, LAS-SNF-XXX-830S; 825 nm, 150 mW), directed from behind and above onto the fly's head and thorax. The laser beam was pulsed (approx. 200 ms pulse width at ∼4 Hz) and its intensity reduced to assure the survival of the fly.

## Learning protocol

Operant self-learning was performed following earlier protocols (*Brembs, 2008*), with timing modifications (first pretest and last test periods were omitted). The direction for straight flight in all experiments at the torque meter was determined as the central value exactly between the maximum left and right turning yaw torque elicited by an optomotor stimulus. The fly's spontaneous yaw torque range was then divided into 'left' and 'right' domains at this value, there is no neutral zone and a 0 value for the torque is very rare but would be considered as positive. After the optomotor stimulus, all visual stimuli were removed from the arena wall, leaving a homogeneously illuminated environment for the fly. During training, heat was applied whenever the fly's yaw torque was in one domain and switched off when the torque passed into the other. Punishment of yaw-torque domains was always set randomly and counterbalanced. In the test phases, heat was permanently switched off and the fly's choice of yaw torque domains recorded.

The experiments are fully automated and computer-controlled. Each fly was used only once. The time-course of each experiment was divided into consecutive periods of 2 min duration. Depending on whether heat was applied during such a period, it was termed a training period (heating possible) or a test period (heat off). Standard experiments consisted of one pre-test period (labeled PI1), four training periods (PI2, PI3, PI5 and PI6) and two memory test periods (PI4, and PI7). At the end of the experiment, the optomotor response was tested again before the fly was challenged by switching the laser on (but pulsing) for about 15 sec. If the optomotor response showed a difference to the values recorded before the experiment, or the fly survived the laser challenge (indicating that the laser was not set properly on the fly's head and that the punishment was not effective), the data were excluded from the analysis.

## Statistical analysis

The yaw torque domain preference of individual flies was quantified as the performance index: $PI = (ta - tb)/(ta + tb)$. During training periods, $tb$ indicates the time the fly is exposed to the heat and $ta$ the time without heat. During tests, $ta$ and $tb$ refer to the times when the fly chose the formerly (or subsequently) unpunished or punished situation, respectively. Thus, a PI of 1 means the fly spent the entire period in the situation not associated with heat, whereas a PI of $-1$ indicates that the fly spent the entire period in the situation associated with heat. Accordingly, a PI of zero indicates that the fly distributed the time evenly between heated and non-heated situations and a PI of 0.5 indicates that 90 of the 120 s in that period were spent in the unpunished situation.

The distributions of PIs during the last test period were presented as boxplots, indicating the median and quartiles with whiskers reaching up to 1.5 times the interquartile range. The superimposed violin plot outlines illustrate kernel probability density, i.e., the width of the shaded area represents the proportion of the data located there.

Code and data were published on Figshare (Labview code for data acquisition (DOI: 10.6084/m9.figshare.664133), R code for data analysis and archiving (DOI: 10.6084/m9.figshare.1561453), masterfile containing the metadata and precomputed data (DOI: 10.6084/m9.figshare.695950) and the raw data grouped by experiments (the

corresponding figshare article number can be read in the masterfile in the "figshareid" column). Rpackages used for the final analysis: cowplot (*Wilke, 2015*), ggplot2 (*Wickham, 2009*), plyr (*Wickham, 2011*), gridExtra (*Auguie, 2015*), rfigshare (*Boettiger et al., 2015*).

## RESULTS

The torque meter apparatus records the yaw torque a fly is producing during stationary flight. Using their optomotor response as a guide, we divided their torque into two domains of similar size. A positive torque domain corresponding roughly to right-turning attempts, and a negative one corresponding to left-turning attempts. During eight minutes of training, either positive or negative yaw torque values were punished by heat. We always report a score representing the proportion of time spent generating the safe behaviour, obtained during a 2-minute test immediately following the last training period. While the score of a single fly can vary from $-1$ to 1, the number of flies with positive scores rises with the progress of the experiment, such that a group of flies is considered capable of learning when the average performance index becomes statistically significantly positive (*Brembs, 2008*).

All PKCs are regulated via a conserved regulatory subunit containing a pseudosubstrate binding the active site. The PKCi method entails overexpressing this pseudosubstrate, which binds to and inhibits every PKC isoform. When expressed ubiquitously, this PKC inhibitor prevents self-learning, without affecting other learning or motor capabilities, or the ability to sense and avoid heat (*Brembs & Plendl, 2008*). In order to elucidate the biochemical processes leading to self-learning, the isoform involved in this process should first be identified. Our first approach was to analyse PKC mutants. At the time, only three lines existed with a homozygous viable mutation for three of the six PKC genes in *Drosophila melanogaster* (*Shieh, Parker & Popescu, 2002*). We outcrossed the mutants to the WTB genetic background and tested these resulting flies, putatively mutant for PKC53e, PKCdelta and PKCinac, respectively. None of these putative mutants was affected in its self-learning ability (Fig. 1A).

In order to avoid putative developmental compensation of a mutated PKC gene by a wild type one, we used temporally controlled RNAi to knock down PKC expression only during adulthood. Because of PKC data from *Aplysia* (*Lorenzetti, Baxter & Byrne, 2008*), we restricted our screen to the Calcium-dependent PKCs, i.e., PKCinaC and PKC53e. We used the TARGET system (*McGuire et al., 2003*), where spatial expression is controlled by a specific driver (the Gal4 line) and temporal control is achieved using a temperature sensitive inhibitor of the GAL4 transcription factor (Gal80$^{ts}$). We expressed Gal4 in all neurons using the elav-Gal4 driver, and Gal80$^{ts}$ in all cells using the tubulin promoter (tub-Gal80$^{ts}$), in combination with the UAS-RNAi constructs. Flies were grown at 25 °C and exposed for two days to 32 °C before being tested. With this regime, the PKC knock-down is supposed to be restricted to neurons during adulthood. The induction of an RNAi construct targeting PKCinaC had no effect on the learning score of the flies, while the insertion of the UAS construct directed against PKC53e both alone and with the driver impaired self-learning (Fig. 1B). Even outcrossing this RNAi line did not restore

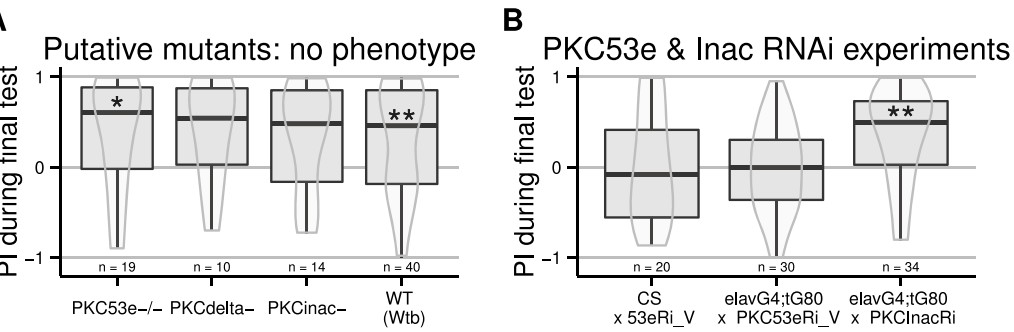

**Figure 1** **Failed attempts to identify the PKC gene involved in operant self-learning.** Performance indices (PI) during a test period following an 8 min training session is reported. LEFT: Flies putatively mutant for PKC genes (PKC-53e, PKC-delta and PCK-InaC) performed well in the self-learning assay. RIGHT: Flies with RNAi constructs targeting PKC53e and PKC InaC were crossed to elav-Gal4;tub-Gal80ts or to CS females. RNAi was induced for two days before the experiment via a 32° heat-shock. While the construct targeting PKC InaC had no effect, the construct for PKC53e prevented self-learning formation even in absence of Gal4 driven expression, such that no firm conclusions can be drawn. Full genotypes of the flies tested are indicated below. CS × 53eRi_V: ;;UAS_PKC53eRNAi_27696/+. elavG4;tG80 × PKC53eRi_V: elavGal4/+;tubGal80ts/+;UAS_PKC53eRNAI_27696/+. elavG4;tG80 x PKCInacRi : elavGal4/+;tubGal80ts/+;UAS_PKCInacRNAI_2895/+. Data is shown as Tukey's boxplots (median is the line surrounded by boxes representing quartiles) with a superposed violinplot. Asterisks indicate significant differences of the scores against 0, using a non-parametrical Wilcox test.

self-learning in the effector line and a second RNAi construct targeting this gene was ineffective (10.6084/m9.figshare.1301620). Hence, in absence of a positive control, we cannot rule out that our heat-shock protocol was ineffective and no conclusions can be drawn from these experiments.

The following experiments were designed to locate the cells in which PKC action is required for self-learning. Again, we used the TARGET system, crossing the tub-Gal80$^{ts}$ and the UAS-PKCi (*Brembs & Plendl, 2008*) constructs into the same fly strain. Flies homozygous for both constructs were then crossed to different Gal4 driver lines to restrict the spatial expression of the PKC inhibitor. At first, we used the pan-neuronal elav-Gal4 driver and tested different heat shock protocols. When the heat-shock was too mild, the inhibition was not effective and the flies were able to learn (Fig. 2A). In contrast, if the temperature was too high, the heat-shock prevented self-learning already in the control flies (Fig. 2B). Using a 4-hour long heat-shock at 35 °C was effective: the control flies showed robust self-learning, while the test flies with inhibition of PKC in all neurons did not: only the flies with the RNAi construct expressed show an indifferent torque distribution during the test phase (Fig. 2C): the group median is close to a score of zero. Note that the sample size of the PKCi control line is low, however the same line was tested in the following experiment and showed no phenotype. We then targeted several central brain areas: the mushroom bodies (a brain area involved in different forms of learning and activity control *Wolf et al. (1998)* and *Brembs (2009)*) with H24-Gal4 and the central complex (a brain region required for a different form of learning performed in the torque meter apparatus *Liu et al. (2006)*) with two lines (c232-Gal4 and a double 7y-Gal4,c819-Gal4).

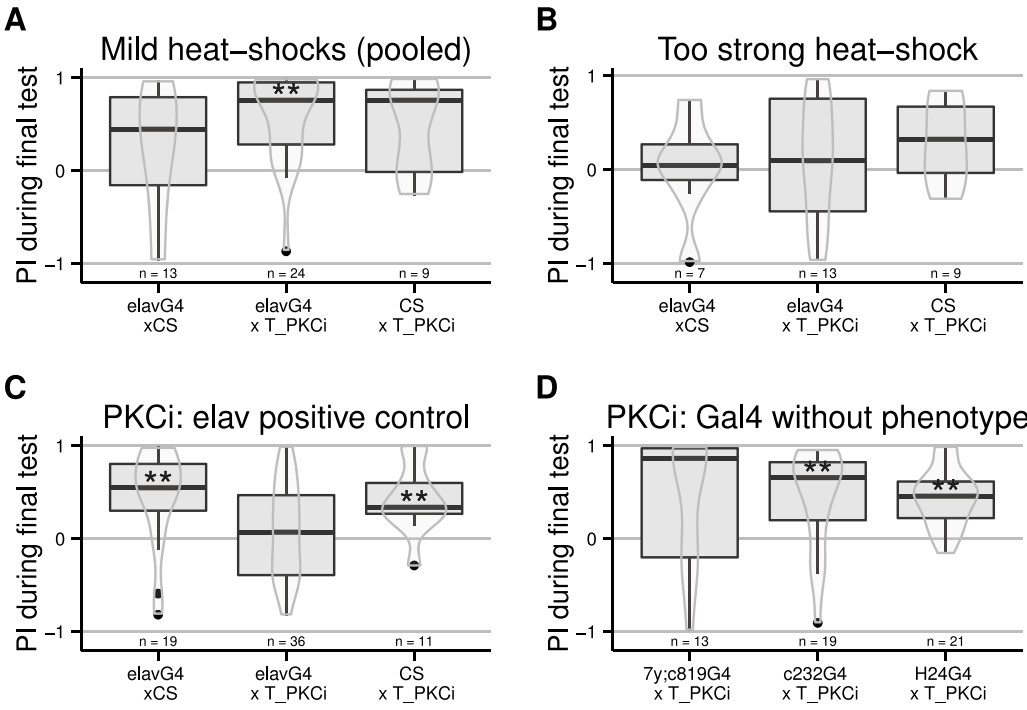

**Figure 2** **PKC inhibition (achieved by using an effective heat-shock protocol) in neurons, but not central brain regions, prevents self-learning formation.** Driving Gal4 in all neurons using the elav-Gal4 driver while inactivating its ubiquitously expressed inhibitor Gal80 with a heat-shock induces the expression of the PKC inhibitor. While test flies are still performing well after a mild heat-shock (A, data pooled from different protocols: 33 ° for 15 h, 36 ° for 2 h, and 37 ° for 1 h), a strong heat-shock prevents learning in control flies (B, 37 ° for 2 h). After a 4-hour heat-shock at 35 °C, test but not control flies were unable to form self-learning (C). Using this latter protocol, we restricted the expression of Gal4 in central brain regions using different drivers targeting central brain regions (D), which were all ineffective in preventing self-learning. Full genotypes of the flies tested are indicated below. elavG4 xCS : elav-Gal4/+. elavG4 × T_PKCi : elav-Gal4/+;tubGal80ts/+; UAS-PKCi/+. CS × T_PKCi : tubGal80ts/+; UAS-PKCi/+. 7y;c819G4 × T_PKCi : tubGal80ts/+; UAS-PKCi/+__ H24-Gal4. c232G4 × T_PKCi : tubGal80ts/+; UAS-PKCi/7y_Gal4,c819-Gal4 . H24G4 × T_PKCi : tubGal80ts/+; UAS-PKCi/c232-Gal4. Data is shown as Tukey's boxplots (median is the line surrounded by boxes representing quartiles) with a superposed violinplot. Asterisks indicate significant differences of the scores against 0, using a non-parametrical Wilcox test.

Flies from these three crosses showed self-learning (Fig. 2D), suggesting that PKC in these central structures is dispensable for self-learning.

Moving away from anatomical structures to classes of neurons, we tested the OK371-Gal4 driver, which drives expression in a majority of glutamatergic neurons (*Mahr & Aberle, 2006*) and d42-Gal4 showing expression in glutamatergic motorneurons (*Parkes et al., 1998*). The induction of the PKCi construct in glutamatergic neurons prevented self-learning, and our data suggest a similar result for the d42 positive neurons (Fig. 3A), although the score in the driver-only control group was not significantly positive. We retested d42-Gal4 along with another Gal4 line known to show expression in motorneurons: c380-Gal4 (*Boerner & Duch, 2010*). Each driver was effective in preventing self-learning when driving PKCi expression, but not without PKCi (Fig. 3B). Finally, we tested lines homozygous for d42-Gal4, for cha-Gal80 (which inhibits the activity of Gal4 in

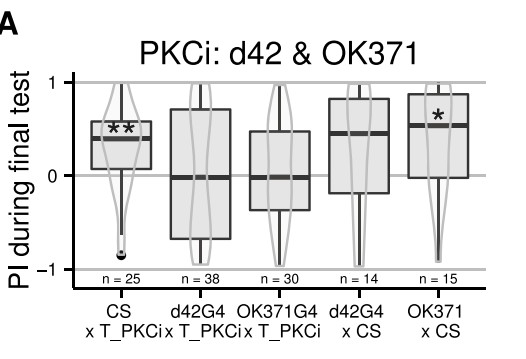

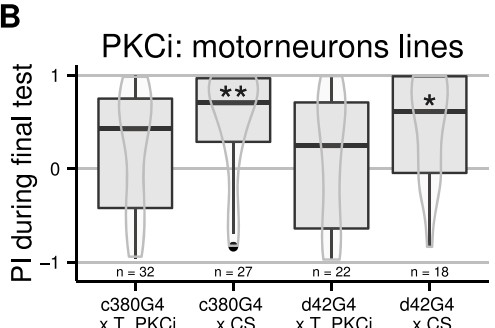

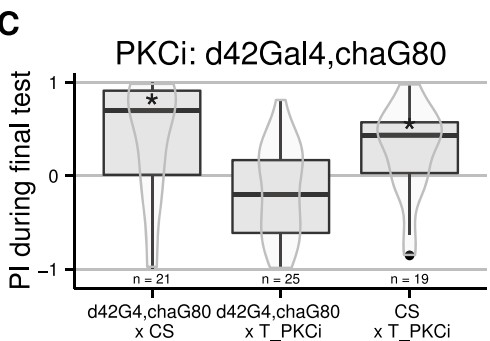

**Figure 3** **Flies with PKCi expression in motorneurons, are impaired in self-learning.** (A) Using OK371-Gal4 (expression in most glutamatergic neurons) or d42-Gal4 to drive PKCi expression was effective in preventing self-learning, while the control flies seem to learn, although the score of the d42Gal4 × CS control line was not statistically significantly positive. (B) Both D42-Gal4 and c380-GAL4 driving PKCi expression overlappingly in motorneurons, yield comparable inhibition of self-learning. (C) The use of the d42Gal4,chaGal80 double construct as a driver was effective in preventing self-learning, while the controls performed well. Heat-shock protocol was a 4-hour heat-shock at 35 °C. Full genotypes of the flies tested are indicated below. CS × T_PKCi: tubGal80ts/+; UAS-PKCi/+. d42G4 × T_PKCi : tubGal80ts/+; UAS-PKCi/d42-Gal4 . OK371G4 × T_PKCi : tubGal80ts/+; UAS-PKCi/+;OK371/+. d42G4 × CS : d42Gal4/+. OK371 × CS : OK371/+.c380G4 × T_PKCi : c380-Gal4/+. c380G4 × CS : c380Gal4/+; tubGal80ts/+; UAS-PKCi/+. d42G4,chaG80 × CS : d42-Gal4, cha-Gal80/+. d42G4,chaG80 × T_PKCi : tubGal80ts/+; UAS-PKCi/d42-Gal4, cha-Gal80 . CS × T_PKCi : tubGal80ts/+; UAS-PKCi/+. Data is shown as Tukey's boxplots (median is the line surrounded by boxes representing quartiles) with a superposed violinplot. Asterisks indicate significant differences of the scores against 0, using a non-parametrical Wilcox test.

cholinergic neurons) and UAS-GFP. These lines did not appear to show any phenotype (10.6084/m9.figshare.1301807), but we suspected that the second UAS construct (UAS GFP) may sequester GAL4, which would reduce the expression of PKCi. Therefore, we outcrossed the d42Gal4,chaGal80 chromosome to get rid of the UAS-GFP construct and tested the flies. The experiment shows that PKC inhibition in non-cholinergic d42Gal4-positive neurons prevents self-learning (Fig. 3C).

## Discussion

In the present work, we have further explored the biochemical basis of self-learning in *Drosphila melanogaster*. Sequence data from several organisms converged on the calcium dependent PKC genes we targeted. Current genetic tools and methods were not sufficient to identify the PKC gene necessary for self-learning (Fig. 1). It is conceivable that due

to functional compensation within the six PKC genes, any mutant approach will be ineffective. On the other hand, our results show the difficulty in using the Target system together with an RNAi knock-down approach. These results suggest that one has to either use a different knock-down strategy and/or monitor PKC expression levels in conjunction with the behavioural screening to be able to draw firm conclusions. While we have not completely exhausted the technical possibilities in this regard, these two failed approaches are instructive for all following experiments.

The TARGET system allowed us to point out motorneurons as one important subset of the neural circuits in which plasticity must occur. Driving PKCi expression in three different lines showing expression in motorneurons blocks self-learning (Figs. 2 and 3). This observation cannot be explained with general learning, motor or sensory deficits in these flies, as expressing PKCi in all cells does not affect any of these general faculties (*Brembs & Plendl, 2008*). Hence, our results suggest that inhibition of PKC in motorneurons is sufficient to prevent operant self-learning in *Drosophila*. However, the expression pattern of the three Gal4 lines is too broad to be able to categorically rule out expression in some non-motorneurons common to all three driver lines. With the largest and most obvious overlap of expression in the motorneurons, we tentatively conclude that self-learning at the torque meter in *Drosophila* requires PKC activity in motorneurons.

Compensatory plasticity throughout the central nervous system allows mammals to walk normally after a change of motorneuron properties following operant reflex conditioning (*Wolpaw, 2010*). Similarly, the optomotor response of flies (measured as yaw torque elicited by an external optomotor stimuli for each fly before and after the experiment, see Materials and Methods) appeared also not to be affected by self-learning. In absence of compensatory plasticity, modifying motorneurons placticity through learning should lead to a shift also in the optomotor response. Interestingly, PKC inhibition in motorneurons (which prevents self-learning formation but should not affect the neurons where compensatory plasticity would occur) had no obvious effect on optomotor response either. This suggests that the compensatory mechanisms do not form in parallel or prior to the PKC-dependent motorneuron plasticity, but in response to it.

In invertebrates, correlates of self-learning have been described in neurons involved in the initiation of a motor program (*Nargeot, Petrissans & Simmers, 2007*), as well as in those involved in deciding which motor program to generate (*Brembs et al., 2002*). In vertebrates and across different animal phyla, motor learning has been found to involve processes along the entire motor control pathway: motor neurons for operant reflex conditioning, the cerebellum for sensory prediction error (as in vestibulo-ocular reflex adaptation), the basal ganglia for learning sequences of actions and the motor cortex for the capacity to execute efficient movements (*Shmuelof & Krakauer, 2011*). Likely, the different brain areas interact during motor learning. Accordingly, we expect motor neurons to be only one of several sites of plasticity underlying self-learning in *Drosophila*.

### Conclusion

We have found evidence that motor-neuron plasticity is involved in self-learning in *Drosophila melanogaster*. This strengthens the hypothesis of a conserved mechanism of

self-learning in all bilaterians (*Shmuelof & Krakauer, 2011*). This is a first and necessary step into our understanding of the neuronal changes at play. The genetic accessibility of this model organism and the convenient experimental design of the assay (*Brembs & Plendl, 2008*) supports the identification and characterization of more genes and neuronal networks playing a role in motor self-learning in the future. Such knowledge would allow us to, for instance, better understand the connection between self-learning and habit formation (*Colomb & Brembs, 2010*), or to design drugs or protocols to optimize motor adaptation in stroke patients. In addition, operant reflex conditioning was shown to have therapeutic potential for spinal patients succeeding in this task (*Thompson, Pomerantz & Wolpaw, 2013*). Unfortunately, the fraction meeting this criterion represents only two thirds of patients (or healthy subjects). It is therefore conceivable that studies on self-learning in invertebrates may lead to discoveries that will help increase this proportion.

An open science approach will help speeding up this research. In this spirit, we included negative results in this publication and adapted our work-flow to make our data and code available upon analysis (DOI: 10.6084/m9.figshare.1561453: while one script pushed the data to Figshare, the R script which produced the figures for this article retrieves the data from there (*Boettiger et al., 2015*)).

## ACKNOWLEDGEMENTS

We want to thank Henrike Scholz, Jean-René Martin, Roland Strauss, Martin Heisenberg, Stephan Sigrist, Carsten Duch as well as the VDRC and the Bloomington Drosophila Stock Center (NIH P40OD018537) for sending fly lines. Special thanks to the Ropensci team and other R developers who made our analysis possible, and to Dr. Tristan Colomb for his help with the labview code.

### Funding

This work was supported by the Swiss National Science Foundation (SNF: PAOOP3_124141) and the Deutsche Forschungsgemeinschaft (DFG:BR 1892/6-1, BR 1892/7-1). The funders had no role in study design, data collection and analysis, decision to publish, or preparation of the manuscript.

### Grant Disclosures

The following grant information was disclosed by the authors:
Swiss National Science Foundation: SNF: PAOOP3_124141.
Deutsche Forschungsgemeinschaft: DFG:BR 1892/6-1, BR 1892/7-1.

### Competing Interests

The authors declare there are no competing interests.

## Author Contributions

- Julien Colomb conceived and designed the experiments, performed the experiments, analyzed the data, contributed reagents/materials/analysis tools, wrote the paper, prepared figures and/or tables, reviewed drafts of the paper.
- Björn Brembs conceived and designed the experiments, contributed reagents/materials/analysis tools, wrote the paper, reviewed drafts of the paper.

## Data Availability

Code and data were published on Figshare (Labview code for data acquisition (DOI: 10.6084/m9.figshare.664133), R code for data analysis and archiving (DOI: 10.6084/m9.figshare.1561453), masterfile containing the metadata and precomputed data (DOI: 10.6084/m9.figshare.695950) and the raw data grouped by experiments (the corresponding figshare article number can be read in the masterfile in the "figshareid" column).

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
