# Peer review of "PKC in motorneurons underlies self-learning, a form of motor learning in Drosophila"

_PeerJ, doi:10.7717/peerj.1971_

## Round 0.1 · original submission · Major Revisions

Dear Authors,

Three reviewers and I have read and appreciated your research. Please address the reviewers' suggestions and revise your manuscript accordingly.

Please use the full species name in all the ms (not just the genus name, Drosophila).

As a side, please note that Drosophila should be typed in italics in the "References" section.

Kind regards,

Reviewer 1 ·

Basic reporting

The author provides about the requirement of PKC in motor neurons for self-learning. The author shows that wild type fly can display self-learning which, restricting their flight direction to avoid punishment and fly with expressed PKCi inhibition fail to build the self-learing. With some clarifications, this manuscript will be a nice example of the use drosophila as a model to study and mapping the important gene and neuron relate to motor learning.

Experimental design

no comments

Validity of the findings

no comments

Additional comments

The author should provide more backgrounds on PKC and PKC pathway involvement in motor neurons plasticity in self-learning.

The author provide evidence to show that most motor neurons are involved. while other part of brain, like Mushroom bodies and central complex are dispensable for self-learning behavior. However, they should provide more background information about why those structure (those gal4 lines) are chosen to be test. For example, what behavior are regulated by MB or central complex? any other evidence showed they are required other motor regulation?


The authors should also more clearly state the significance of their finding of PKC activity level in motor neurons are required in self-learning and how their work adds on to what is known about the system in flies and other mammals. ( how does understanding this in flies extend or corroborate what is known in mammals or human? )

·

Basic reporting

The article is very well written and the data are very clearly presented.

Experimental design

Excellent.

Validity of the findings

Excellent.

Additional comments

In thus study the authors explored the role of different PKC genes in self-learning in Drosophila. Previous studies have shown that a general PKC inhibitor expressed ubiquitously prevents self-learning, but here the authors show that disruption in any individual PKC gene was not sufficient to prevent learning. This suggests redundancy or the ability of the system to compensate for loss of one gene. This is an interesting and important finding for the field. In addition by restricting temporally and spatially the expression of the general PKC inhibitor the authors show that PKC is required in motor neurons. This provides an important step towards elucidating the neural substrates of self-learning in Drosophila.

Reviewer 3 ·

Basic reporting

No comments

Experimental design

No comments

Validity of the findings

No comments

Additional comments

I red the manuscript on the biochemical basis of operant self-leraning in Drosophila with real interest.The manuscript is generally well written and introduction is clear and exhaustive. Methods fit with the porpouse of the study and results are appropriately discussed. However I detected some criticisms listed below.
LN 30 Avoid the use double round brackets, replacing the external ones with square brackets. Please change it all over the text.
LN 31, LN 44, (and all over the text) Please list the references cited as requested by PeerJ Author Guidelines
LN 47 Provide full names before use abbreviations (e.g. PKC)
LNs 68-70 This sentence seems not appropriate in introduction section. You should move it to conclusions.
LN 85 Tested flies were allowed to contact any other conspecific before been glued to the hook? Which was the mating status of tested females?
LN 112-115 I suppose that author considered yaw torque to take place when flies did not fly straight. Did you consider as a yaw torque also small flight deviation (I was thinking about deviation of few degrees) or did you use a range to identify straight flights?
LNs 122-123 Surviving tests at the end of the experiment do not sound really ethical. Why so important to assess survival to laser challenge?
FIGURES The number of replicates are very low for some treatments, in particular in Figure2. In my opinion, comparing a treatment with 11 replicates with a different one constituted by 36 rep could be misled or at least they should be appropriately discussed. In addition, I found the Figures not self-explained. Indeed, is difficult to understand the exact meaning of figure components. To improve readability, should be added information about each component, such as the meaning of the black horizontal line, the gray areas presented over and under, the black vertical lines, and lastly the curve gray lines.

---

## Round 0.2 · accepted · Accept

Dear Authors,

The comments from the three Reviewers have been properly addressed.

Although only one of the original reviewers opted to re-review your revision they were satisfied with the revision, and they confirmed that the comments of the other reviews were also addressed. Based on their input, as well as on my own reading of your responses, I am pleased to accept your manuscript for publication in PeerJ.

Kind regards,

Reviewer 3 ·

Basic reporting

OK

Experimental design

OK

Validity of the findings

OK

Additional comments

I found this manuscript really improved after previous revisions. Since the authors positively took into account all suggestions and comments of the reviewers, in my opinion this article should be accepted as it is.